

# Free fermions beyond Jordan and Wigner

Paul Fendley[1] and Balázs Pozsgay[2]

**1** All Souls College and Rudolf Peierls Centre for Theoretical Physics,
University of Oxford, Parks Rd, Oxford OX1 3PU, United Kingdom
**2** MTA-ELTE "Momentum" Integrable Quantum Dynamics Research Group,
Department of Theoretical Physics, ELTE Eötvös Loránd University, Budapest, Hungary

## Abstract

The Jordan-Wigner transformation is frequently utilised to rewrite quantum spin chains in terms of fermionic operators. When the resulting Hamiltonian is bilinear in these fermions, i.e. the fermions are free, the exact spectrum follows from the eigenvalues of a matrix whose size grows only linearly with the volume of the system. However, several Hamiltonians that do not admit a Jordan-Wigner transformation to fermion bilinears still have the same type of free-fermion spectra. The spectra of such "free fermions in disguise" models can be found exactly by an intricate but explicit construction of the raising and lowering operators. We generalise the methods further to find a family of such spin chains. We compute the exact spectrum, and generalise an elegant graph-theory construction. We also explain how this family admits an $N = 2$ lattice supersymmetry.



# 1   Intro

Integrable models by their nature are special: they possess a hierarchy of commuting charges that strongly constrain the physical behaviour. A consequence is that sometimes certain types of computations yield exact results. An array of powerful techniques, including the Bethe ansatz and Yang-Baxterology, have been developed to do these calculations. Yet even when these tools are applicable, important physical properties defy description. For example, even though the Bethe equations contain enough information to determine the full spectrum, in practice it is usually possible only to obtain exact energies only for the ground state and some low-lying states, and even then only in the limit of large system size.

A notable exception to the latter statement comes from free-fermion systems, where one can compute the exact spectrum directly in a number of models of great interest. The method utilised is called a Jordan-Wigner transformation [1], and the output is a Hamiltonian or transfer matrix written in terms of fermion bilinears. One then can use now-standard techniques [2] to compute the eigenvalues of such a free-fermion Hamiltonian, yielding

$$E = \pm\epsilon_1 \pm \epsilon_2 \pm \cdots \pm \epsilon_M \,, \tag{1}$$

where the energy levels $\epsilon_k$, $k = 1, 2, \dots M$ are eigenvalues of a single matrix, and so are all roots of a single polynomial. A key feature of this free-fermion spectrum is that choice of the $\pm$ signs giving the $2^M$ energies does not affect the values of the $\epsilon_k$.

The Jordan-Wigner procedure has been utilised thousands of times over the last century. Nevertheless, only recently was a method developed to determine whether a given Hamiltonian can be transformed to fermion bilinears in such fashion [3] without using trial-and-error (see also [4]). Even more remarkably, the method is graphical. To any Hamiltonian acting on two-state systems one associates a *frustration graph*, and only if this graph satisfies certain properties can the Jordan-Wigner transformation be used to rewrite the Hamiltonian in terms of fermion bilinears.

It is natural to ask if all known Hamiltonians with spectrum (1) admit a Jordan-Wigner transformation to fermion bilinears. The answer is no, and even before [3] several counterexamples were known. The first (that we know of) is the "Cooper-pair" chain, where with particular open boundary conditions, a Bethe-ansatz computation yields the spectrum (1) [5]. Each state here exhibits large degeneracies, but changing to the seemingly simpler periodic boundary conditions splits the degeneracies and wrecks the free-fermion behaviour. Another interesting feature of the Cooper-pair chain is that it exhibits a $N = 2$ lattice supersymmetry [6,7] where the Hamiltonian can be written as a square of either of two supersymmetry generators. Subsequently, the Cooper-pair model with open boundary conditions was shown to be equivalent (via a unitary transformation) [8] to another fermion chain with lattice supersymmetry, the "DFNR" model [9].

Another model with spectrum (1) not solvable by a Jordan-Wigner transformation was found in [10] and dubbed "free fermions in disguise" (ffd). The Hilbert space is a chain of

two-state systems, and the Hamiltonians for periodic and open boundary conditions are given in terms of Pauli matrices as

$$H_{\text{ffd,p}} = \sum_{j=1}^{L+1} r_j X_j X_{j+1} Z_{j+2}, \qquad H_{\text{ffd,o}} = \sum_{j=1}^{L-1} r_j X_j X_{j+1} Z_{j+2}, \qquad (2)$$

respectively, where the indices in the former but not the latter are interpreted mod $L+1$. A Jordan-Wigner transformation gives a Hamiltonian comprised entirely of four-fermion operators. The parity-conjugate model with the terms $Z_{j-1} X_j X_{j+1}$ is obviously equivalent, but less obviously commutes with (2).

The ffd model has no $U(1)$ symmetry, so the Bethe ansatz is at best extremely difficult to implement. The proof that $H_{\text{ffd,o}}$ has spectrum of the form (1) is thus very different from that of the Cooper-pair/DFNR chain. Instead, the raising and lowering operators were constructed explicitly and directly by a tractable but rather intricate method [10]. As we describe in detail below, the construction involves defining a transfer matrix with nice properties, and the method yields a solution of the corresponding two-dimensional classical model as well. There are $2^M$ different energies in (1), while there are $2^L$ eigenstates of $H$. Each energy here has exponentially large degeneracies, because $M$ grows linearly in $L$ but with $M/L < 1$. The detailed analysis shows that the degeneracies are identical for each choice of $\pm$ signs.

Remarkably, there exists a simple graphical procedure to determine if a model can be solved by the method of [10] [11]. When the frustration graph obeys certain properties (it is "claw-free" and "even-hole" free), the Hamiltonian has spectrum (1). A feature of these methods [10–12] is that integrability can be established without need of Yang-Baxter equation, even in the presence of the spatially varying couplings. Further many-body Hamiltonians solvable by this method were found in [12,13] and the two graphical procedures were put in a common structure [12]. However, the frustration graph for the Cooper-pair/DFNR model does not satisfy the needed properties, so its free-fermion spectrum does not follow automatically.

Nonetheless, the Cooper-pair/DFNR and ffd models exhibit a number of properties in common beyond the free-fermion spectrum, including the requirement of certain open boundary conditions, an extended supersymmetry algebra, and extensive degeneracies. Moreover, rewriting the DFNR Hamiltonian in terms of spins makes it simpler and the resemblance closer:

$$H_{\text{DFNR,f}} = Z_0 Z_2 + \sum_{j=1}^{L-1} \left( X_j X_{j+1} Z_{j+2} + Z_{j-1} Y_j Y_{j+1} + Z_j Z_{j+2} \right). \qquad (3)$$

This Hamiltonian acts on $L+2$ two-state systems labeled $0, 1, \ldots L+1$, and we impose fixed boundary conditions where the edge spins are eigenstates of $Z_0$ and $Z_{L+1}$.

In this paper we find and analyse an integrable model interpolating between the ffd and DFNR models, deriving its free-fermion spectrum for fixed boundary conditions. For spatially uniform couplings, its Hamiltonian is

$$H_b = b Z_0 Z_2 + \sum_{j=1}^{L-1} \left( X_j X_{j+1} Z_{j+2} + b^2 Z_{j-1} Y_j Y_{j+1} + b Z_j Z_{j+2} \right), \qquad (4)$$

so that $H_0 = H_{\text{ffd,o}}$ and $H_1 = H_{\text{DFNR,f}}$ indeed. The core of our proof is that of [10,11], but we need to extend the construction beyond its original applicability. At the end we find all the energy levels $\epsilon_k$ via the roots of a single polynomial, recovering (1).

In section 2, we detail the Hamiltonian, its supersymmetry, and the algebra its generators satisfy. We also outline how we found it by exploiting the "medium-range" integrability approach of [14]. By constructing explicitly a series of commuting conserved quantities, we show in section 3 that the model is integrable for both periodic and fixed boundary conditions,

even for spatial varying couplings. We also derive an extended (super) symmetry algebra valid only for fixed boundary conditions. The transfer matrix and its inversion relation are derived in section 4. Finally, in section 5 we construct the raising/lowering operators explicitly and derive the exact free-fermion spectrum for fixed boundary conditions.

## 2 The Hamiltonian and its supersymmetry

### 2.1 Medium-range integrability

We start by explaining how we found the particular combination of couplings in (4). Since below we give a detailed analysis of the more general problem using very different techniques, we here simply outline the procedure.

Yang-Baxter integrable models have a transfer matrix $t(u)$ constructed from local Lax operators so that $[t(u), t(u')] = 0$. In the standard case of nearest-neighbour interactions, the Lax operator is constructed from a solution of the Yang-Baxter equation called the $R$ matrix [15]. The integrable Hamiltonian is the first logarithmic derivative

$$\mathcal{H} = \partial_u \log(t(u))|_{u=0}, \tag{5}$$

with local conserved charges arising from higher derivatives.

For spin chains with longer (but finite) interaction ranges, the framework was extended in [14]. Commuting transfer matrices were found by enlarging the auxiliary spaces and by choosing $R$-matrices that factorise into products of Lax operators at selected points in rapidity space. These ideas yield an integrability condition generalising the Reshetikhin condition [16] for nearest-neighbour interactions. This condition can be used to construct and classify integrable models. For recent applications of this idea in other settings see for example [17–19].

The medium-range integrability condition works as follows. For simplicity we consider only three-site interactions, writing $\mathcal{H} = \sum_j g_j$, where $g_j$ is a local operator spanning three sites. Then we construct an extensive operator

$$\mathcal{H}_2 = \sum_j \left( \left[ g_j, g_{j+1} + g_{j+2} \right] + \tilde{g}_j \right), \tag{6}$$

where $\tilde{g}_j$ is another three-site operator. If the Hamiltonian is derived via (5), and the Lax operator satsifes a certain regularity condition, then $\mathcal{H}_2$ is the next local charge obtained from $t(u)$, i.e. in finite volume

$$\left[ \mathcal{H}, \mathcal{H}_2 \right] = 0. \tag{7}$$

Note that the form of $\mathcal{H}_2$ is very restrictive: its operator density is a five-site operator dominated by the commutator, as the arbitrary term $\tilde{g}_j$ spans only three sites.

One often can reverse the logic to find an integrable Hamiltonian. Namely, one treats (7) as a cubic equation for the Hamiltonian density $g_j$ (and linear in $\tilde{g}_j$). If one finds a solution for any system size, then there should exist an $R$-matrix which solves the Yang-Baxter relation yielding commuting transfer matrices and a Lax operator. While not proved, for all known solutions of (7) they do indeed exist. The same is true here. Taking $\mathcal{H}$ to be the periodic DFNR Hamiltonian but allowing arbitrary (but spatially homogenous) coefficients for each of the three types of the terms, a direct computation then shows that (7) is satisfied if and only if $\mathcal{H} = H_{b,\mathrm{p}}$, where [20]

$$H_{b,\mathrm{p}} = \sum_{j=1}^{L+1} \left( X_j X_{j+1} Z_{j+2} + b^2 Z_{j-1} Y_j Y_{j+1} + b Z_{j-1} Z_{j+1} \right), \tag{8}$$

and we interpret indices mod $L+1$ and so identify sites 0 and $L+1$. The corresponding Lax pair then indeed can be found, ensuring integrability of $H_{b,p}$.

## 2.2 The supersymmetry generator

The nicest way to define the Hamiltonian with open boundary conditions and spatially varying couplings is in terms of supersymmetry generators. Our model thus provides an example of many-body supersymmetric quantum mechanics [6,7].

The supersymmetry operators are non-local in terms of the spins, and the most transparent way of writing supersymmetry generators is in terms of Majorana-fermion operators defined as

$$\psi_{2j-1} = X_j \prod_{k=0}^{j-1} Z_j, \qquad \psi_{2j} = i Z_j \psi_{2j-1}. \tag{9}$$

These operators obey $\{\psi_k, \psi_{k'}\} = 2\delta_{kk'}$. The building blocks of the first supersymmetry generator are trilinears in the Majorana operators, namely

$$\alpha_j = i a_j \psi_{2j} \psi_{2j+1} \psi_{2j+2} = a_j \psi_{2j} Z_{j+1}, \qquad \beta_j = i b_j \psi_{2j-3} \psi_{2j-2} \psi_{2j} = b_j \psi_{2j} Z_{j-1}, \tag{10}$$

for arbitrary real numbers $a_j$ and $b_j$. In discussing supersymmetry, we restrict to open boundary conditions. The supersymmetry generator is defined as

$$Q = \frac{i}{\sqrt{2}} \sum_{j=1}^{L} (\alpha_j + \beta_j) = \frac{1}{\sqrt{2}} \sum_{j=1}^{L} \psi_{2j} (a_j Z_{j+1} + b_j Z_{j-1}). \tag{11}$$

The supersymmetric Hamiltonian is then simply the square of $Q$:

$$H = Q^2 - \frac{1}{2} \sum_{j=1}^{L} (a_j^2 + b_j^2), \tag{12}$$

where we subtract off the constant piece. Since $Q$ anticommutes with the fermion-parity symmetry

$$(-1)^F = \prod_{j=0}^{L+1} (i \psi_{2j-1} \psi_{2j}) = \prod_{j=0}^{L+1} Z_j, \tag{13}$$

it maps between equal-energy states in even and odd sectors under $(-1)^F$, except for states annihilated by it. Since $Q$ is Hermitian for real $a_j$ and $b_j$, (12) requires that the energy is bounded from below by the (negative) constant and that any state annihilated by $Q$ must be a ground state.

## 2.3 The Hamiltonian

To give an explicit expression for $H$, we utilise the algebra of the building blocks

$$\{\alpha_j, \beta_{j'}\} = 0, \text{ for } j \neq j', \qquad \{\alpha_j, \alpha_{j'}\} = \{\beta_j, \beta_{j'}\} = 0, \text{ for } |j - j'| > 1,$$
$$(\alpha_j)^2 = a_j^2, \qquad (\beta_j)^2 = b_j^2, \qquad [\alpha_{j-1}, \alpha_j] = [\beta_{j-1}, \beta_j] = [\alpha_j, \beta_j] = 0. \tag{14}$$

Any pair of terms that anticommutes cancels in $Q^2$, so the Hamiltonian is comprised of products of any two commuting building blocks, namely

$$A_j \equiv \alpha_{j-1} \alpha_j = a_{j-1} a_j X_{j-1} X_j Z_{j+1}, \qquad B_j \equiv \beta_{j-1} \beta_j = b_{j-1} b_j Z_{j-2} Y_{j-1} Y_j,$$
$$C_j \equiv \alpha_j \beta_j = a_j b_j Z_{j-1} Z_{j+1}. \tag{15}$$

For fixed boundary conditions we set $a_0 = b_0 = a_{L+1} = b_{L+1} = 0$ and require the edge spins to be eigenstates of $Z_0$ and $Z_{L+1}$. Using (12) then yields the Hamiltonian

$$H = C_1 + \sum_{j=2}^{L} (A_j + B_j + C_j), \tag{16}$$

generalising (4) to spatially varying couplings. The periodic version is

$$H_{\mathrm{p}} = \sum_{j=1}^{L+1} (A_j + B_j + C_j), \tag{17}$$

with indices identified mod $(L+1)$. The uniform periodic Hamiltonian (8) is recovered from (17) by setting $a_j = 1$ and $b_j = b$ for all $j$.

The rest of the paper is spent analysing $H$ and $H_{\mathrm{p}}$, showing both are integrable and the former has a free-fermion spectrum. A useful tool is a graphical visualisation of the algebra (14), where each $\alpha_j$ and $\beta_j$ corresponds to a vertex on the square ladder, with adjacent vertices corresponding to generators that *commute*. For fixed boundary conditions, the graph is

$$\begin{array}{ccccccccc}
\alpha_1 & \overset{A_2}{\rule{1.5em}{0.4pt}} & \alpha_2 & \overset{A_3}{\rule{1.5em}{0.4pt}} & \alpha_3 \rule{1em}{0.4pt} & & \rule{1em}{0.4pt}\ \alpha_{L-1} & \overset{A_L}{\rule{1.5em}{0.4pt}} & \alpha_L \\
C_1 \big| & & C_2 \big| & & C_3 \big| & \cdots & C_{L-1} \big| & & C_L \big| \\
\beta_1 \rule{1.5em}{0.4pt} & \underset{B_2}{} & \beta_2 \rule{1.5em}{0.4pt} & \underset{B_3}{} & \beta_3 \rule{1em}{0.4pt} & & \rule{1em}{0.4pt}\ \beta_{L-1} & \underset{B_L}{\rule{1.5em}{0.4pt}} & \beta_L
\end{array} \tag{18}$$

To avoid cluttering the picture, we omit the edges connecting the nodes to themselves. Each term $A_j$, $B_j$, $C_j$ in the Hamiltonian then corresponds to an *edge* of this square ladder. We thus often refer to each such term as a *dimer*. For periodic boundary conditions, the graph is

$$\begin{array}{ccccccccccc}
\overset{A_0}{\rule{1.5em}{0.4pt}} & \alpha_0 & \overset{A_1}{\rule{1.5em}{0.4pt}} & \alpha_1 & \overset{A_2}{\rule{1.5em}{0.4pt}} & \alpha_2 & \overset{A_3}{\rule{1.5em}{0.4pt}} & \alpha_3 \rule{1em}{0.4pt} & & \rule{1em}{0.4pt}\ \alpha_{L-1} & \overset{A_L}{\rule{1.5em}{0.4pt}}\ \alpha_L \rule{0.5em}{0.4pt} \\
& C_0 \big| & & C_1 \big| & & C_2 \big| & & C_3 \big| & \cdots & C_{L-1} \big| & C_L \big| \\
\underset{B_0}{\rule{1.5em}{0.4pt}} & \beta_0 & \underset{B_1}{\rule{1.5em}{0.4pt}} & \beta_1 & \underset{B_2}{\rule{1.5em}{0.4pt}} & \beta_2 & \underset{B_3}{\rule{1.5em}{0.4pt}} & \beta_3 \rule{1em}{0.4pt} & & \rule{1em}{0.4pt}\ \beta_{L-1} & \underset{B_L}{\rule{1.5em}{0.4pt}}\ \beta_L \rule{0.5em}{0.4pt}
\end{array} \tag{19}$$

where the lines at the left and right ends are joined.

A key relation shows that the dimer operators are not all independent of one other. Instead,

$$A_j B_j = \alpha_{j-1} \alpha_j \beta_{j-1} \beta_j = -\alpha_{j-1} \beta_{j-1} \alpha_j \beta_j = -C_{j-1} C_j \quad \implies \quad B_j = -\frac{1}{a_{j-1}^2 a_j^2} A_j C_{j-1} C_j. \tag{20}$$

The $B_j$ thus can be written as a non-linear product of the others. In terms of the pictures (18) and (19), the corresponding dimers obey

$$\begin{array}{ccc}
\alpha_{j-1} \overset{A_j}{\rule{1.5em}{0.4pt}} \alpha_j & & \alpha_{j-1} \cdots\cdots \alpha_j \\
\vdots \qquad\qquad \vdots & = & - \quad C_{j-1} \big| \qquad C_j \big| \\
\beta_{j-1} \underset{B_j}{\rule{1.5em}{0.4pt}} \beta_j & & \beta_{j-1} \cdots\cdots \beta_j
\end{array}. \tag{21}$$

Just as the algebra (14) is built from fermionic operators, we can build an algebra from the independent dimer operators $A_j$ and $C_j$. Their algebra is easy to work out using the pictures: two operators anticommute if either (i) the corresponding dimers share a single vertex, or (ii) a single edge touches both dimers. Thus we have

$$\{A_j, A_{j+1}\} = \{A_j, A_{j+2}\} = \{A_j, C_{j-2}\} = \{A_j, C_{j-1}\} = \{A_j, C_j\} = \{A_j, C_{j+1}\} = 0. \tag{22}$$

All other pairs of $A_j$ and $C_{j'}$ commute. Those involving the $B_j$ can be found by using (21). Because of the $\alpha_j \leftrightarrow \beta_j$ symmetry of the algebra (vertical reflection of the ladder), one can replace $A$ with $B$ in (22), along with

$$\{A_j, B_{j-1}\} = \{A_j, B_{j+1}\} = 0. \tag{23}$$

All other pairs of dimer operators commute, in particular $[A_j, B_j] = [C_j, C_{j'}] = 0$.

## 2.4 A second supersymmetry generator

The DFNR Hamiltonian from (3) has a $U(1) \times U(1)$ symmetry and a second supersymmetry generator [9]. The latter but not the former holds for the full model (16), but with an interesting caveat. At $a_j = b_j = 1$ the other generator can be found by commuting with the $U(1)$ charges

$$\mathcal{Q}_{\text{odd}} = \sum_{j \text{ odd}} Z_j Z_{j+1}, \qquad \mathcal{Q}_{\text{even}} = \sum_{j \text{ even}} Z_j Z_{j+1}. \tag{24}$$

Using $[Z_j, \psi_{2k}] = 2i\delta_{jk}\psi_{2j-1}$ gives

$$\tfrac{1}{2}\big[\mathcal{Q}_{\text{odd}}, Q\big]_{a_j=b_j=1} = \tfrac{1}{2}\big[\mathcal{Q}_{\text{even}}, Q\big]_{a_j=b_j=1} = \frac{i}{\sqrt{2}} \sum_{j=0}^{L} \psi_{2j-1}\big(1 + Z_{j-1}Z_{j+1}\big). \tag{25}$$

Squaring this operator does indeed yield $H_{DFNR}$ plus a constant. Using this hint it is straightforward to find another supersymmetry generator for any couplings $a_j$, $b_j$, but only for fixed boundary conditions:

$$\widetilde{Q} = \frac{1}{\sqrt{2}} \sum_{j=1}^{L} \left[ \psi_{2j-1}\left( a_j \prod_{k=1}^{j} \frac{b_k}{a_k} + b_j Z_{j-1}Z_{j+1} \prod_{k=1}^{j} \frac{a_k}{b_k} \right) \right]. \tag{26}$$

It satisfies $H = \widetilde{Q}^2 + \text{const}$ and so commutes with $H$. The constant here in general is different from that in (12). For $a_j$ and $b_j$ independent of $j$, the constant is larger and so the lower bound on $H$ (arising from the non-negative eigenvalues of $Q^2$) is not as strong. The two supersymmetry generators obey $\{Q, \widetilde{Q}\} = 0$, so that $H$ has (at least) $N = 2$ supersymmetry for any couplings. However, away from the DFNR point neither $Q$ nor $\widetilde{Q}$ annihilates any states, so all states form doublets under each of the supersymmetries.

# 3 The conserved charges

We here show here that the Hamiltonians $H$ and $H_{\text{p}}$ with fixed and periodic boundary conditions are integrable. We do so by constructing explicitly a sequence of commuting conserved charges. These charges are not purely of academic interest – knowing them explicitly is essential to our subsequent calculation of the spectrum. We cannot however immediately apply the conserved-charge construction of [10] because this model does *not* satisfy the necessary graphical conditions of [11, 12]. We here first review the earlier procedure, and then explain how to generalise it to our Hamiltonians. The relation (21) between the generators proves absolutely essential.

## 3.1 Conserved charges from graphs

The first step is to write the Hamiltonian as a sum of terms $h_m$ with the following properties:

$$H = \sum_{m=1}^{M} h_m, \qquad (h_m)^2 = r_m^2, \quad \text{either } \big[h_m, h_n\big] = 0, \text{ or } \{h_m, h_n\} = 0, \quad \forall\, m, n, \tag{27}$$

for $r_m$ a real number. Our Hamiltonians have

$$h_{3j-2} = C_j, \qquad h_{3j-4} = B_j, \qquad h_{3j-3} = A_j, \qquad (28)$$

with $M = 3L - 2$ and $3L + 3$ for fixed and periodic boundary conditions respectively.

Such interactions can be encoded in a frustration graph. This graph $G$ has a vertex for each $m$ and an edge between $m$ and $n$ when $\{h_m, h_n\} = 0$ [11]. The graph is easy to draw in the special case $H_{\text{ffd,o}}$ from (2), which comes from setting $\beta_j = 0$ in $H$, leaving only the $A_j$. Since $\{A_j, A_{j+1}\} = \{A_j, A_{j+2}\} = 0$ with other pairs commuting, the resulting $G$ for open boundary conditions is a zig-zag ladder:

$$G_{\text{ffd}} = \qquad \qquad \cdots \qquad\qquad (29)$$

For our general Hamiltonians (16, 17) the graph is rather nasty-looking, and it is more convenient to work with the ladder (18). Nonetheless, we will show how a modified frustration graph plays an essential role in our analysis.

The original construction of conserved charges for the ffd chain was by explicit computation [10]. For $H_{\text{ffd,o}}$, the first non-trivial charge is

$$Q^{(3)} = \sum_{j < k-2 < l-4} A_j A_k A_l, \qquad (30)$$

i.e. is the sum over of all products of three different generators that commute with other. The remarkable result of [11] was that this construction of conserved charges follows solely from properties of the frustration graph, and so applies to a more general class of models. A commuting sequence of non-trivial conserved charges occurs when the frustration graph is *claw-free*. A claw is conveniently displayed via an "induced" subgraph. An induced subgraph is defined by specifying a subset of vertices of the graph and then including all edges connecting those vertices. A claw corresponds to a quartet of operators $a, b, c, d \in \{h_m\}$ whose induced subgraph is

$$(31)$$

The operators thus obey

$$[b,c] = [b,d] = [c,d] = \{a,b\} = \{a,c\} = \{a,d\} = 0. \qquad (32)$$

The frustration graph for the ffd chain drawn in (29) is claw-free.

For a claw-free frustration graph, the conserved quantities are given in terms of independent sets. An independent subset of a graph is a collection of vertices such that no pair shares an edge. Let $G$ be the frustration graph associated with a Hamiltonian obeying (27), so that each independent subset corresponds to a collection of $r$ operators $h_m$ such that each pair commutes with each other. Let $S^{(r)}$ be the set of all independent subsets of $G$ with $r$ vertices. Then a theorem of [11] shows that for claw-free $G$, all charges

$$Q^{(r)} = \sum_{S \in S^{(r)}} \prod_{s \in S} h_s, \qquad (33)$$

are conserved, i.e. $[H, Q^{(r)}] = 0$. Each term on the right-hand side of (33) corresponds to an induced subgraph of $r$ disconnected vertices. The ordering in the product does not matter, as by definition all the $h_s$ commute when the $s$ belong to an independent set. This expression indeed reduces to (30) for $H_{\text{ffd,o}}$, as apparent from the corresponding frustration graph (29).

The reason for the claw-free condition is straightforward to understand. First note that in $H^2$, pairs that anticommute necessarily vanish, leaving only independent sets and constants:

$$H^2 = \sum_{m,m'} h_m h_{m'} = \sum_{m,m' \big| [h_m, h_{m'}]=0} h_m h_{m'} = 2Q^{(2)} + \sum_m r_m^2. \tag{34}$$

Obviously, $H^2$ commutes with $H$, so $Q^{(2)}$ trivially commutes with $H$. The cube is more interesting. When the claw-free condition applies, $H^3$ can be split into two pieces that *individually* commute with the Hamiltonian. Consider a term $bcd$ in $H^3$ with $b \neq c \neq d \neq b$. Two types of such term remain in the sum: the "non-local" ones where $[b,c] = [b,d] = [c,d] = 0$, and the "local" ones where $\{b,c\} = \{c,d\} = [b,d] = 0$. The corresponding induced subgraphs are

$$\text{non-local:} \quad \overset{\bullet}{b} \quad \overset{\bullet}{c} \quad \overset{\bullet}{d} \,, \qquad \text{local:} \quad \overset{\bullet}{b}\!\!-\!\!\overset{\bullet}{c}\!\!-\!\!\overset{\bullet}{d}\,. \tag{35}$$

The only other terms remaining in $H^3$ are those where two or all of the three operators coincide.

The non-local terms (35) in $H^3$ are all independent subsets in $S^{(3)}$, so summing over all of them gives $Q^{(3)}$. Now consider $[H, Q^{(3)}]$. The only terms potentially remaining in this commutator are those where each term $h_m$ of $H$ either anticommutes with precisely one of $b, c, d$ or anticommutes with all three of them. The latter case, however, cannot occur when the frustration graph does not have a claw (32). The terms in $[H, Q^{(3)}]$ coming from the non-local terms in $Q^{(3)}$ thus correspond to an induced subgraph of the form

$$
\begin{array}{c}
\overset{h_m}{\bullet} \\
b\bullet\!\!\diagup \quad \overset{\bullet}{c} \quad \bullet d
\end{array}
\tag{36}
$$

since $h_m \neq b, c, d$. The key observation is such an induced subgraph cannot arise from commuting any $h_m$ with any of the other terms in $H^3$, neither the local ones from (35) nor those where operators coincide. The sum over the terms of the form (36) therefore must vanish in order for $H^3$ to commute with $H$, yielding $[H, Q^{(3)}] = 0$.

The proof that all $Q^{(r)}$ defined by (33) commute with $H = Q^{(1)}$ for claw-free $G$ is simple [11]. Consider the commutator $[h_m, \prod_{s \in S} h_s]$ between one term in $H$ and one in $Q^{(r)}$. This commutator is non-vanishing if $h_m$ anticommutes with an odd number of the $h_s$ in the product. When $G$ is claw-free, the only possibility for a non-vanishing commutator is for $h_m$ to anticommute with only a single one of the $h_s$, which we label $h_{s'}$. We then consider the subset $S_m \subset S$ comprised of the elements of $S$ other than $s'$, i.e. those that commute with $m$. Another independent subset of $G$ is therefore $m \cup S_m$, and we have

$$\left[ h_m + h_{s'}, \prod_{s \in S} h_s + \prod_{s \in m \cup S_m} h_s \right] = \left[ h_m + h_{s'}, \left( h_{s'} + h_m \right) \prod_{s \in S_m} h_s \right] = 0. \tag{37}$$

All terms in the commutator $\left[ H, Q^{(r)} \right]$ cancel two by two in this fashion, and so this commutator vanishes for claw-free $G$.

More work is required to show that all $Q^{(r)}$ defined by (33) mutually commute for claw-free $G$, and we defer it to section 3.3.

## 3.2 Conserved charges for periodic and fixed boundary conditions.

The construction of conserved charges from the procedure of [10–12] does not immediately apply to our Hamiltonians $H$ and $H_p$. The reason is that there are claws (32) in the frustration graph, for example from taking $a = A_j$ or $B_j$ and then $b, c, d$ to be any three of

$C_{j-2}$, $C_{j-1}$, $C_j$, $C_{j+1}$. The arguments of the preceding section 3.1 no longer automatically apply.

All hope is not lost: we can exploit the fact not all terms in the Hamiltonian are independent, as shown in (21). First consider $Q^{(2)}$, which as seen in (34) is the sum over all products of two commuting generators. It thus includes the combination $A_j B_j + C_{j-1} C_j = 0$, so that all such "forbidden pairs" $A_j B_j$ and $C_{j-1} C_j$ cancel in the sum in $Q^{(2)}$. This observation suggests we redefine $Q^{(r)}$ to exclude forbidden pairs from any product. We thus construct a modified frustration graph $\overline{G} \supset G$ by adding edges to $G$ between $C_j$ and $C_{j+1}$ and between $A_j$ and $B_j$ for all $j$. Then we define $\overline{Q}^{(r)}$ in the same fashion as (33) but instead using the modified frustration graph $\overline{G}$:

$$\overline{Q}^{(r)} = \sum_{\overline{S} \in \overline{\mathcal{S}}^{(r)}} \prod_{\overline{s} \in \overline{S}} h_{\overline{s}} . \tag{38}$$

Because forbidden pairs cancel in the sum, $\overline{Q}^{(2)} = Q^{(2)}$ here. The operators for higher values of $r$, however, are different. Because of restriction to independent sets, the highest value of $r$ allowed for non-zero couplings is $r_{\max} = \lfloor (L+1)/2 \rfloor$.

The definition (38) amounts to removing all the terms in $Q^{(r)}$ from (33) that involve any of the forbidden pairs. In any term, the generators can be no closer than

$$A_j B_{j\pm 2}, \quad C_j C_{j+2}, \quad A_j C_{j+2}, \quad B_j C_{j+2}, \quad A_j C_{j-3}, \quad B_j C_{j-3}, \quad A_j A_{j+3}, \quad B_j B_{j+3} . \tag{39}$$

Examples of terms in $\overline{Q}^{(3)}$ for $L = 6$ and fixed boundary conditions are

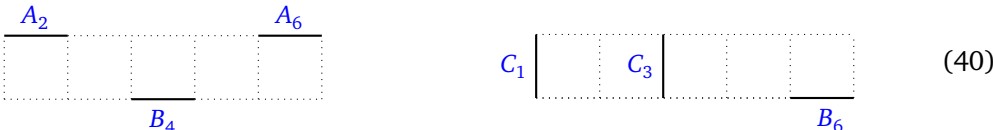

$$\tag{40}$$

For periodic boundary conditions at $L + 1 = 6$, $\overline{Q}_{\mathrm{p}}^{(3)} = C_1 C_3 C_5 + C_2 C_4 C_6$.

Here we extend the proof of section 3.1 to show that $\overline{Q}^{(r)}$ commutes with the Hamiltonian. It is straightforward to check that all the claws of $G$ involve one of the forbidden pairs, so that they do not appear in $\overline{G}$. Consider $[h_m, \prod_{\overline{s} \in \overline{S}} h_{\overline{s}}]$ for some independent set $\overline{S} \in \overline{G}$, and denote by $h_{m'}$ the other part of the forbidden pair involving $h_m$. By construction of $\overline{G}$, the situation in (32) never occurs in this commutator, leaving only contributions where $h_m$ anticommutes with a single term $h_{\overline{s}'}$ in the product. If $m' \notin \overline{S}$, contributions to $[H, \overline{Q}^{(r)}]$ cancel two by two by splitting $\overline{S} = \overline{s}' \cup \overline{S}_m$ and then proceeding as in (37). However, when $m' \in \overline{S}$, the argument no longer works because $h_m h_{m'}$ cannot be included in any product, i.e. $m \cup \overline{S}_m$ is not an independent set of $\overline{G}$.

Such terms in the commutator cancel in a different way. Because of (21), each forbidden pair $(h_m, h_{m'})$ partners with another forbidden pair $(h_{m''}, h_{m'''})$. As pictured, the four operators correspond to dimers on a single square, with the two comprising a forbidden pair on opposite sides from each other. They obey

$$h_m h_{m'} = -h_{m''} h_{m'''}, \qquad \{h_{m'''}, h_{m'}\} = \{h_m, h_{m''}\} = [h_{m'''}, h_{m''}] = [h_m, h_{m'}] = 0 . \tag{41}$$

For the case at hand with $m' \in \overline{S}$, we have by assumption

$$\{h_m, h_{s'}\} = [h_{m'}, h_{s'}] = 0 . \tag{42}$$

Then necessarily one of $(h_{m''}, h_{m'''})$ commutes with $h_{s'}$ and the other anticommutes, say

$$\{h_{m'''}, h_{s'}\} = [h_{m''}, h_{s'}] = 0 . \tag{43}$$

This relation follows either by using the decomposition (15) or simply by checking all cases explicitly. For example, consider $h_m = A_j$ so that $h_{m'} = B_j$ and the only possibilities for $h_{s'}$ satisfying (42) are $A_{j\pm 2}$. Then for $h_{s'} = A_{j+2}$, we have $h_{m''} = C_{j-1}$ and $h_{m'''} = C_j$, while for $h_{s'} = A_{j-2}$, we have $h_{m''} = C_j$ and $h_{m'''} = C_{j-1}$. In terms of pictures, these two examples are



$$\tag{44}$$

where we use dashed lines for the dimer operators coming from $\overline{S}'$. We define $\overline{S}'_m$ by removing $m'$ as well as $s'$ from $\overline{S}$. Because of (43), the set $(m'', s') \cup \overline{S}'_m$ is independent, and so appears in $\overline{Q}^{(r)}$. We then have

$$\left[h_m + h_{m'''}, \prod_{\bar{s}\in\overline{S}} h_{\bar{s}} + \sum_{\bar{s}\in(m'',s')\cup S'_m} h_{\bar{s}}\right] = \left[h_m + h_{m'''}, \left(h_{m'} + h_{m''}\right)h_{s'}\right]\prod_{\bar{s}\in S'_m} h_s = 0, \tag{45}$$

where we use the essential relation (21) along with (42, 43) to show that

$$\left[h_m, h_{m'}h_{s'}\right] + \left[h_{m'''}, h_{m''}h_{s'}\right] = \left[h_m, h_{m''}h_{s'}\right] = \left[h_{m'''}, h_{m'}h_{s'}\right] = 0. \tag{46}$$

Thus commutators involving forbidden pairs also cancel in the sums.

Nowhere in these arguments was it necessary to invoke the fixed boundary conditions of $H$; we only needed the algebra (22) along with the condition (21) (or equivalently the algebra (14) and the definitions (15)). Thus if we define the charges $\overline{Q}^{(r)}$ and $\overline{Q}_{\rm p}^{(r)}$ from (38) using the graphs $\overline{G}$ for fixed and periodic boundary conditions respectively, they are conserved:

$$\left[H, \overline{Q}^{(r)}\right] = \left[H_{\rm p}, \overline{Q}_{\rm p}^{(r)}\right] = 0. \tag{47}$$

### 3.3 Commuting charges

Here we extend the proof of [11] to show that the conserved charges commute amongst themselves:

$$\left[\overline{Q}^{(r)}, \overline{Q}^{(r')}\right] = 0. \tag{48}$$

A single term in this commutator can be written as

$$\mathcal{C}_{\overline{S}\,\overline{S}'} = \left[\prod_{l=1}^{k} h_{s_l}, \prod_{l'=1}^{k'} h_{s'_{l'}}\right], \tag{49}$$

where we have denoted the two independent sets in the charges as $\overline{S} = (s_1, s_2, \dots s_k) \in \overline{S}^{(r)}$ and $\overline{S}' = (s'_1, s'_2, \dots s'_{k'}) \in \overline{S}^{(r')}$. As with the proof of conserved charges, we show that non-vanishing terms in the commutator cancel two by two in the sum over all subsets.

When no forbidden pairs appear in the product in $\mathcal{C}_{\overline{S}\,\overline{S}'}$, we can use the proof of Lemma 1 of [11]. The first step is to decompose $\mathcal{C}_{\overline{S}\,\overline{S}'}$ into a product over *paths* on the frustration graph. An open path $\mathcal{P} = (p_1, p_2, \dots, p_{|\mathcal{P}|})$ has the property that each operator in the corresponding list $(h_{p_1}, h_{p_2}, \dots, h_{p_{|\mathcal{P}|}})$ anticommutes with its successor and predecessor, and commutes with the rest:

$$\{h_{p_i}, h_{p_{i+1}}\} = 0, \qquad [h_{p_i}, h_{p_{i'}}] = 0, \quad \text{for } |i - i'| > 1. \tag{50}$$

In a closed path the indices are interpreted cyclically mod $|\mathcal{P}|$. The *length* of the path is the number of vanishing anticommutators, i.e. the number of pairs of non-commuting operators. For a closed path, the length is the number of operators $|\mathcal{P}|$, while for an open path the length is $|\mathcal{P}|-1$.

A term $h_{s_l}$ that commutes with all the $h_{s'_{l'}}$ forms a path with a single operator and so is of length zero. Longer paths correspond to operators alternating between the two charges

$$\left(h_{s_i}, h_{s'_{i'}}, h_{s_{i+1}}, h_{s'_{i'+1}}, \dots\right), \tag{51}$$

with the corresponding induced subgraph of $G$ for an odd-length open path being

For an odd-length path, there are the same number of $s_i$ and $s'_{i'}$, and the corresponding commutator

$$\mathcal{C}_{\overline{S}\cap\mathcal{P},\overline{S}'\cap\mathcal{P}} = \left[\prod_{l=i}^{i-1+|\mathcal{P}|/2} h_{s_l}, \prod_{l'=i'}^{i'-1+|\mathcal{P}|/2} h_{s_{l'}}\right], \tag{53}$$

is non-vanishing, while the anticommutator of the same two operators vanishes. For example, $[A_2 C_4, B_3 A_6] = -2A_2 B_3 C_4 A_6$ corresponds to a path of length three pictured by

The reverse is true for even-length paths: the corresponding commutator vanishes, while the anticommutator does not.

The only non-vanishing terms in the commutator of charges are therefore those $\mathcal{C}_{\overline{S}\,\overline{S}'}$ that possess an odd number of odd-length open paths. (A closed path of the form (51) is by necessity even-length.) Pick one of these odd-length open paths $\mathcal{P}$. Since by construction all terms in (51) commute with all other operators in $\overline{S}'$ and $\overline{S}$ not part of the path, we can define two new independent sets $\widetilde{S}, \widetilde{S}'$ by swapping all the operators $h_{s_i} \in \overline{S} \cap \mathcal{P}$ in the path with those $h_{s'_{i'}} \in \overline{S}' \cap \mathcal{P}$. This is the generalisation of the swap we did in (37); that case corresponds to a path length of one. Because $\mathcal{C}_{\overline{S}\cap\mathcal{P},\overline{S}'\cap\mathcal{P}} + \mathcal{C}_{\overline{S}'\cap\mathcal{P},\overline{S}\cap\mathcal{P}} = 0$ by definition, the same crucial cancellation happens here:

$$\mathcal{C}_{\overline{S}\,\overline{S}'} + \mathcal{C}_{\widetilde{S}\,\widetilde{S}'} = 0. \tag{55}$$

When an open path of the form (51) is of odd length, the total number of operators $|\mathcal{P}|$ is even, as apparent in the picture (52). Thus the swap does not change the number of operators in each product so that $\widetilde{S} \in \overline{S}^{(r)}$ and $\widetilde{S}' \in \overline{S}^{(r')}$. The cancellation (55) therefore means that all terms in (48) involving no forbidden pairs cancel two by two.

Terms involving forbidden pairs cancel by a generalisation of the argument leading to (45). Since they do not appear in $\overline{Q}^{(r)}$, the only way a forbidden pair $(h_m, h_{m'})$ can appear in (49) is when one $m \in \overline{S}$ and $m' \in \overline{S}'$. Moreover, the only way the presence of a forbidden pair can change matters is if it interferes with the swap needed for the cancellation in (55). Thus we only need consider forbidden pairs involving an odd-length path.

The needed cancellations depend on how the forbidden pair appears. There are few enough cases so that we just go through them one by one. It is convenient to display the cases using induced subgraphs as in (52) with a forbidden pair (i.e. an edge in $\overline{G}$ but not $G$)

denoted by a dotted line. The two configurations in the cancellation in (45) then have induced subgraphs

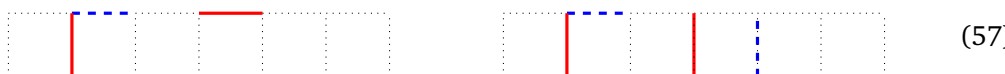 (56)

The same cancellation obviously happens in the case of any path with the forbidden pair remaining on the end.

Thus we now need only consider cases where the forbidden pair appears in the middle of a path, or joins two paths. There are two types of such joints. One is similar to (56), and so we again exploit (45) and the identity (21) with no need of swapping. The joints of this type in the cancelling cases pictured as e.g.

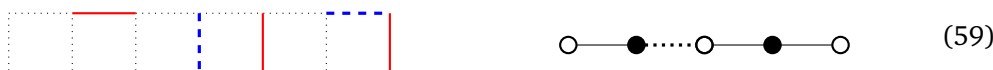 (57)

Extending (56), the corresponding induced subgraphs are the same but with changed labels:

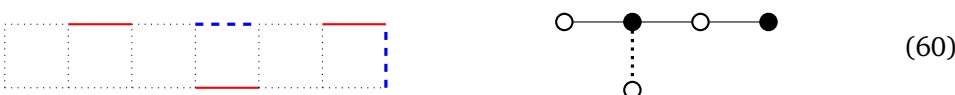 (58)

Since the operators away from the joint do not change between the two configurations, the cancellation happens for any length of paths on either side.

A subtler type of joint can occur with the forbidden pair $C_{j-1}C_j$, e.g.

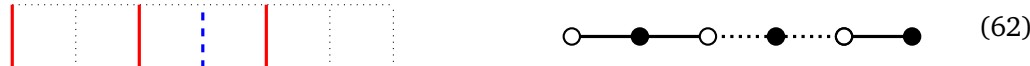 (59)

Here we cannot use (21) directly without any swap. Instead, the cancelling configuration does not have a joint, but rather a forbidden pair $A_j B_j$ in the middle of an odd-length path. For example, cancelling (59) is

(60)

Such an odd-length path must have an even number of operators on one side of the forbidden pair, and odd on the other. There are the same number of operators from $\overline{S}$ and from $\overline{S}'$ on the even side. We thus swap only these operators between $Q^{(r)}$ and $Q^{(r')}$, so that the number of operators in each stays the same. To check that the signs work out properly, (59) relates to (60) via

$$\left(A_3 B_5 A_7\right)\left(A_5 C_7\right) = -A_3 B_5 A_5 A_7 C_7 = A_3 C_4 C_5 A_7 C_7 = -\left(A_3 C_5 C_7\right)\left(C_4 A_7\right), \quad (61)$$

as needed for the cancellation. Again, the paths on either end can be extended. As long as the total path length (not including dotted lines) remains odd, one side always has an even number of operators and edges. That side swaps without extra signs, e.g. extending the path by including one additional operator on each side gives

$$\left(A_3 B_5 A_7 B_9\right)\left(A_2 A_5 C_7\right) = -A_3 B_5 A_5 A_7 B_9 A_2 C_7 = A_3 C_4 C_5 A_7 B_9 A_2 C_7 = -\left(A_2 C_4 A_7 B_9\right)\left(A_3 C_5 C_7\right).$$

It makes no difference if other forbidden pairs occur; if they are on the even side they just swap along with the others.

The remaining cases to check are when multiple forbidden pairs are adjacent, e.g.

(62)

Here and whenever the number of operators in the forbidden pairs connecting the two paths is odd, the number of operators of each type is the same whenever the total path length is odd. Thus one can simply swap between the two sets to get the desired cancellation. When the number of operators in the forbidden pairs is even, e.g.

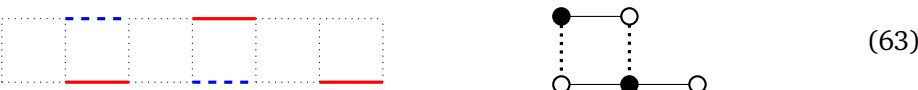

(63)

the cancellation happens differently. Instead, this configuration pairs and cancels with

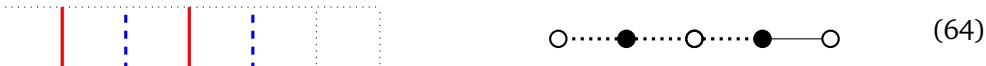

(64)

The signs here are those needed for the cancellation:

$$\left(B_3 A_5 B_7\right)\left(A_3 B_5\right) = B_3 A_3 A_5 B_5 B_7 = C_2 C_3 C_4 C_5 B_7 = -\left(C_2 C_4 B_7\right)\left(C_3 C_5\right).$$

(65)

The extension to more forbidden pairs in the joint is straightforward. Moreover, lengthening the paths away from the forbidden pairs is possible via the same arguments as before, as long as the total path length (not including dotted lines) remains odd. We thus have established (48).

## 3.4 More conserved charges for fixed boundary conditions

All of the results for conserved charges above apply to both fixed and periodic boundary conditions. However, the former possesses more symmetry, just as in the $b_j = 0$ limit [10]. A first inkling of why fixed boundary conditions are special is apparent in the second supersymmetry generator $\widetilde{Q}$ from (26), as it exists for $L \to \infty$ in general only for fixed boundary conditions. Here we show how $H$ (but not $H_{\mathrm{p}}$) possesses a entire hierarchy of symmetries beyond supersymmetry. A consequence is the extensive degeneracies in the free-fermion spectrum.

The additional conserved charges for fixed boundary conditions are built from non-local combinations of the building blocks $\alpha_j$, $\beta_j$ of the supersymmetry generator $Q$. We relabel them as

$$\gamma_j = \begin{cases} \alpha_j, & j \text{ odd}, \\ \beta_j, & j \text{ even}, \end{cases} \qquad \delta_j = \begin{cases} \alpha_j, & j \text{ even}, \\ \beta_j, & j \text{ odd}. \end{cases}$$

(66)

These operators obey

$$\left[\gamma_j, \delta_j\right] = \left[\gamma_j, \delta_{j\pm1}\right] = 0, \qquad \{\gamma_j, \gamma_{j'}\} = \{\delta_j, \delta_{j'}\} = \{\gamma_k, \delta_{k'}\} = 0, \text{ for } |k - k'| > 1,$$

(67)

and for all $j, j'$. The supersymmetry generator can be split as

$$Q = Q_\gamma + Q_\delta, \qquad Q_\gamma = \sum_{j=1}^L \gamma_j, \quad Q_\delta = \sum_{j=1}^L \delta_j.$$

(68)

Each operator $Q_\gamma$ and $Q_\delta$ squares to a constant and so commutes with the Hamiltonian individually:

$$H = \{Q_\gamma, Q_\delta\}, \qquad \left[Q_\gamma, H\right] = \left[Q_\delta, H\right] = 0.$$

(69)

The extra symmetry generators constructed in [10] also generalise to our model. To proceed, we first consider the commutator $[Q_\gamma, Q_\delta]$, which obviously commutes with $H$ as a consequence of (69). Less obviously, we can split it into two pieces as with the supercharge $Q$:

$$\tfrac{1}{2}\left[Q_\gamma, Q_\delta\right] = Q_{\gamma\delta} - Q_{\delta\gamma}, \qquad Q_{\gamma\delta} = \sum_{j \leq k-2} \gamma_j \delta_k, \quad Q_{\delta\gamma} = \sum_{j \leq k-2} \delta_j \gamma_k,$$

(70)

each of which commutes with $H$ individually. Since the operators in $Q_{\gamma\delta}$ are separated like the $h_m$ are in $\overline{Q}^{(r)}$, the proof is similar. First note that

$$H = \sum_{i=1}^{L} \gamma_i \sum_{l=0,\pm1} \delta_{i+l} = \sum_{i=1}^{L} \delta_i \sum_{l=0,\pm1} \gamma_{i+l}, \tag{71}$$

where for fixed boundary conditions $\delta_0 = \gamma_0 = \delta_{L+1} = \gamma_{L+1} = 0$. As in (37), non-vanishing contributions to the commutator $[H, Q_{\gamma\delta}]$ cancel two by two. Such contributions $\gamma_i \delta_{i+l} \gamma_j \delta_k$ must have either $i + l = j$, $j \pm 1$, or $i = k$, $k \pm 1$. More specifically, non-vanishing terms of the first type require $(i, i+l, j, k)$ to be one of

$$(j-2, j-1, j, k), (j-1, j-1, j, k), (j-1, j, j, k), (i, i, i-1, k), (i, i-1, i-1, k), (i, i-1, i-2, k),$$

where $k \geq j + 2$. Without this additional constraint on $k$ in the latter three the corresponding commutator vanish. For each term of this type, the term with $i \leftrightarrow j$ also appears in the list and has opposite sign, so that the two cancel in the sum over all. For example,

$$\gamma_{j-2}\delta_{j-1}\gamma_j\delta_k = -\gamma_j\delta_{j-1}\gamma_{j-2}\delta_k, \qquad \gamma_{j-1}\delta_{j-1}\gamma_j\delta_k = -\gamma_j\delta_{j-1}\gamma_{j-1}\delta_k.$$

Terms of the second type cancel two-by-two in the same fashion. From (70) and (69), it follows that $[H, Q_{\delta\gamma}] = 0$ as well. The fixed boundary conditions are essential, as the splitting makes no sense for periodic.

A series of conserved charges follows by this construction. The operators

$$\begin{aligned}
Q_{\gamma\delta\gamma} &= \tfrac{1}{2}\{Q_\gamma, Q_{\gamma\delta}\} = \tfrac{1}{2}\{Q_\gamma, Q_{\delta\gamma}\} = \sum_{j \leq k-2 \leq j'-4} \gamma_j \delta_k \gamma_{j'}, \\
Q_{\delta\gamma\delta} &= \tfrac{1}{2}\{Q_\delta, Q_{\gamma\delta}\} = \tfrac{1}{2}\{Q_\delta, Q_{\delta\gamma}\} = \sum_{j \leq k-2 \leq j'-4} \delta_j \gamma_k \delta_{j'},
\end{aligned} \tag{72}$$

automatically commute with $H$. The terms must alternate between $\delta$ and $\gamma$ in this fashion because the other contributions to the anticommutator cancel two by two as above. Non-trivial conserved charges do follow by using the same commuting and splitting procedure from (70)

$$\tfrac{1}{2}[Q_\gamma, Q_{\delta\gamma\delta}] = -\tfrac{1}{2}[Q_\delta, Q_{\gamma\delta\gamma}] = Q_{\gamma\delta\gamma\delta} - Q_{\delta\gamma\delta\gamma},$$
$$Q_{\gamma\delta\gamma\delta} = \sum_{j \leq k-2 \leq j'-4 \leq k'-6} \gamma_j \delta_k \gamma_{j'} \delta_{k'}, \qquad Q_{\delta\gamma\delta\gamma} = \sum_{j \leq k-2 \leq j'-4 \leq k'-6} \delta_j \gamma_k \delta_{j'} \gamma_{k'}. \tag{73}$$

The proof that $Q_{\gamma\delta\gamma\delta}$ commutes with $H$ is essentially the same as for $Q_{\gamma\delta}$.

Continuing in this fashion yields a series of non-trivial and non-local bosonic commuting charges $Q_{\gamma\delta\gamma\delta...}$. A crucial distinction with the $\overline{Q}^{(r)}$ is that these charges do *not* in general commute amongst one another. Their full algebra, however, is not known (at least not by us). Obviously, it would be interesting to find it, or even to find a commuting subalgebra. Such a large non-abelian symmetry algebra implies that the energy levels are highly degenerate, a fact we prove in section 5.

## 4 The transfer matrix and local commuting quantities

The next step in the derivation of the spectrum is to construct a one-parameter family of commuting transfer matrices, and show they satisfy a nice inversion relation. They generate local conserved quantities, despite the longer-range interactions. In this section we describe these steps, restricting to fixed boundary conditions here and for the remainder of the paper.

Commuting transfer matrices are comprised of a series of the commuting conserved quantities:

$$T(u) = \sum_{r=0}^{r_{max}} (-u)^r \overline{Q}^{(r)} \quad \implies \quad [T(u), T(u')] = 0, \tag{74}$$

with $\overline{Q}^{(0)} = 1$, $\overline{Q}^{(1)} = H$ and $r_{max} = \lfloor (L+1)/2 \rfloor$. It is useful below to have a recursion relation for $T(u)$. To write it out, we include a subscript $T_L$ for the transfer matrix acting on the $L$-site Hilbert space i.e. $\alpha_j = \beta_j = 0$ for $j > L$ and $j < 1$. We also define $T_L^A$ such that $\beta_L = 0$ as well, and $T_L^B$ such that $\alpha_L = 0$ instead. These have the effect of shortening one of the legs of the ladder on which the dimers live. Then the transfer matrices for varying sizes obey the recursion relations

$$\begin{aligned} T_L &= T_L^A + T_L^B - T_{L-1} - uC_L T_{L-2}, \\ T_{L+1}^A &= T_L - uA_{L+1}T_{L-1}^B, \qquad T_{L+1}^B = T_L - uB_{L+1}T_{L-1}^A, \end{aligned} \tag{75}$$

for $L > 0$, defining $T_0 = T_{-1} = T_0^A = T_1^A = T_0^B = T_1^B = 1$.

## 4.1 Inversion relation

For fixed boundary conditions, $T(u)$ satisfies a very nice inversion relation, namely

$$T(u)T(-u) = P(u^2), \tag{76}$$

where the polynomial $P(x)$ is built from independent sets in the same fashion as the charges

$$P(x) = \sum_{r=0}^{r_{max}} (-x)^r \sum_{\overline{S} \in \overline{S}^{(r)}} \prod_{\overline{s} \in \overline{S}} (r_{\overline{s}})^2. \tag{77}$$

To prove (76), we show that all the terms in $T(u)T(-u)$ cancel save the "diagonal" ones. Writing out the product in terms of the charges gives

$$T(u)T(-u) = \sum_{r'=0}^{r_{max}} \sum_{r=0}^{r_{max}} (-1)^r u^{r+r'} \overline{Q}^{(r)} \overline{Q}^{(r')} = \sum_{l=0}^{r_{max}} \sum_{r=0}^{l} (-1)^r u^{2l} \overline{Q}^{(r)} \overline{Q}^{(2l-r)}. \tag{78}$$

As reflected in the last expression, the terms with $r + r'$ odd cancel because of the $(-1)^r$ and the fact that the charges commute. We see already from (76) and the fact that $Q^{(2)} = \overline{Q}^{(2)}$ that the terms with $r + r' = 2$ indeed sum to $-u^2 \sum_m (r_m)^2$. The "diagonal" terms then give a polynomial constructed from independent sets in the same way as the charges, so that relations

$$\begin{aligned} P_L &= P_L^A + P_L^B - P_{L-1} - (ua_L b_L)^2 P_{L-2}, \\ P_{L+1}^A &= P_L - (ua_L a_{L+1})^2 P_{L-1}^B, \qquad P_{L+1}^B = P_L - (ub_L b_{L+1})^2 P_{L-1}^A, \end{aligned} \tag{79}$$

hold for $L > 0$, defining $P_0 = P_{-1} = P_0^A = P_1^A = P_0^B = P_1^B = 1$.

Following [11], we again decompose each term in (78) into a products of paths. Expanding out the charges in terms of independent sets $\overline{S}$ and $\overline{S}'$ as in (49) gives each term as

$$\mathcal{A}_{\overline{S}\overline{S}'} = \prod_{l=1}^{r} h_{s_l} \prod_{l'=1}^{r'} h_{s'_{l'}}. \tag{80}$$

Since the commutator (53) for an odd-length path without forbidden pairs is non-vanishing, the corresponding anticommutator vanishes. Thus any time such a path appears in $\mathcal{A}_{\overline{S}\overline{S}'}$, we

can define new configurations $\widetilde{S}$, $\widetilde{S}'$ by swapping the operators within the path, analogously to (55). The swap does not change $r$ or $r'$, so

$$\mathcal{A}_{\overline{S}\cap\mathcal{P},\overline{S}'\cap\mathcal{P}} + \mathcal{A}_{\overline{S}'\cap\mathcal{P},\overline{S}\cap\mathcal{P}} = 0 \quad \Longrightarrow \quad \mathcal{A}_{\overline{S}\,\overline{S}'} + \mathcal{A}_{\widetilde{S}\widetilde{S}'} = 0. \tag{81}$$

When forbidden pairs are included, we then can rerun all the arguments from section 3.3 to define the cancelling configuration for any odd-length path, including those with forbidden pairs.

Thus the only potentially non-vanishing contributions to (78) come from $\mathcal{A}_{\overline{S}\,\overline{S}'}$ with only even-length paths, or those diagonal terms with $\overline{S} = \overline{S}'$. The factor $(-1)^r$ is crucial to the cancellations of the former. To illustrate, consider a zero-length path for some term in (80), i.e. an $h_{s_i}$ that commutes (but does not form a forbidden pair) with all $h_{s'_{i'}}$, We then can define independent sets $\widetilde{S}$ and $\widetilde{S}'$ by moving this $s_i$ from $\overline{S}$ to $\overline{S}'$. As $\widetilde{S} = \overline{S}/s_i$ has one less element and $\widetilde{S}' = s_i \cup \overline{S}'$ one more, $\mathcal{A}_{\widetilde{S}\widetilde{S}'}$ appears in the product $\overline{Q}^{(r-1)}\overline{Q}^{(r'+1)}$. Because $\mathcal{A}_{\overline{S}\,\overline{S}'} = \mathcal{A}_{\widetilde{S}\widetilde{S}'}$ here, the $(-1)^r$ in (78) ensures that these two terms cancel in the sum over $r$ and $r'$.

Such a swap is possible for any even-length path without forbidden pairs, and so all such paths do not contribute. We thus need to extend the proof of [11] to allow for forbidden pairs in even-length paths. The cancellations discussed in section 3.3 that do not require a swap (i.e. only utilise (21)) apply here as well. Thus the arguments given above in (56) or (57, 58) also mean the analogous even-length paths also cancel two by two.

Requiring more work is the case where a forbidden pair occurs in the middle of an even-length path, analogous to (59) or (60)). Since the full path is of even length, there must be either an odd or an even number of operators on each of the sides of the forbidden pair. The latter possibility cancels as before, since the swap described in (59, 60) only occurs on the even-length side, independently of what happens on the other side. When both sides are odd, e.g.

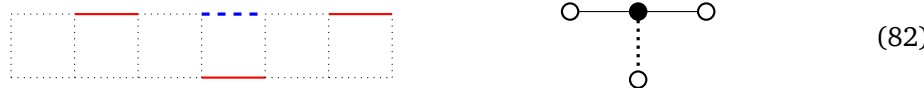

$$\tag{82}$$

the cancelling operator is then

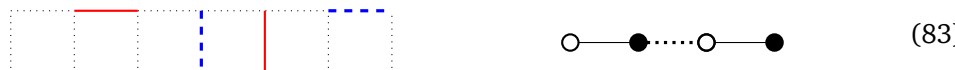

$$\tag{83}$$

Here $r$ increases by one while $r'$ decreases. The swap here does not result in a sign change, so that $\mathcal{A}_{\overline{S}\,\overline{S}'} = \mathcal{A}_{\widetilde{S}\widetilde{S}'}$. The above example corresponds to

$$\left(A_3 B_5 A_7\right)\left(A_5\right) = -A_3 B_5 A_5 A_7 = A_3 C_4 C_5 A_7 = \left(A_3 C_5\right)\left(C_4 A_7\right). \tag{84}$$

Since the swap increases the number of operators in one charge by one and decreases the other by one, the cancellation comes from the $(-1)^r$ in (78). Again, the paths on either end can be extended with the same arguments as above.

The final case to check is that with multiple adjacent forbidden pairs, as for example

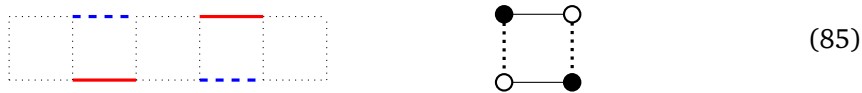

$$\tag{85}$$

This configuration pairs and cancels with

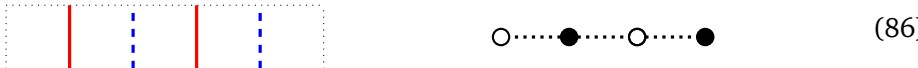

$$\tag{86}$$

The signs here amount to

$$\left(B_3 A_5\right)\left(A_3 B_5\right) = -B_3 A_3 A_5 B_5 = -C_2 C_3 C_4 C_5 = -\left(C_2 C_4\right)\left(C_3 C_5\right), \tag{87}$$

as needed for the cancellation. Extending this calculation to longer paths is straightforward, e.g.

$$\left(B_3 A_5 B_7\right)\left(A_3 B_5 C_8\right) = B_3 A_3 A_5 B_5 B_7 C_8 = C_2 C_3 C_4 C_5 B_7 C_8 = -\left(C_2 C_4 B_7\right)\left(C_3 C_5 C_8\right), \tag{88}$$

gives the needed sign for cancellation.

The only terms that contribute to (78) are those without any paths at all. These are the "diagonal" terms, where $\overline{S} = \overline{S}'$. Summing over all of them with the weight $u^{r+r'}$ gives the polynomial $P(u^2)$ defined in (77). We thus have proved (76).

## 4.2 Local commuting quantities

We then can construct a family of local conserved quantities from the logarithmic derivative $-\frac{\partial}{\partial u} \ln T(u)$. To order $u^2$,

$$
\begin{aligned}
-\frac{1}{P(u^2)} T'(u) T(-u) &= \left(1 + u^2 \sum_m r_m^2 + \dots\right)\left(H - 2u\overline{Q}^{(2)} + 3u^2 \overline{Q}^{(3)}\right)\left(1 + uH + u^2 \overline{Q}^{(2)}\right) \\
&= H + u\left(H^2 - 2\overline{Q}^{(2)}\right) + u^2\left(3\overline{Q}^{(3)} - H\overline{Q}^{(2)} + H\sum_m r_m^2\right) + \dots \\
&= H + uH^{(2)} + u^2 H^{(3)} - \dots
\end{aligned}
$$

Since all the non-vanishing terms in $H^2$ are comprised of pairs of operators that commute with each other (see (34)), the only remaining terms in $H^{(2)}$ are constants, i.e.

$$H^{(2)} = H^2 - 2\overline{Q}^{(2)} = \sum_m r_m^2. \tag{89}$$

The first non-trivial local conserved charge is therefore

$$-H^{(3)} = H\overline{Q}^{(2)} - 3\overline{Q}^{(3)} - H\sum_m r_m^2. \tag{90}$$

The subtractions remove all non-local terms in $H\overline{Q}^{(2)}$, leaving $H^{(3)}$ local.

# 5 The raising/lowering operators

The crucial property needed for a free-fermion spectrum is the existence of the raising and lowering operators satisfying a Clifford algebra. We construct them in this section, utilising an algebraic relation between the transfer matrix and an edge mode in the same fashion as [10]. We then show that combining this relation with the commuting charges and the inversion relation (76) allows the Hamiltonian to be written as a bilinear in the raising and lowering operators. The free-fermion spectrum then follows directly.

## 5.1 The nice algebraic relation

Key to our construction is a *edge operator* $\alpha_{L+1}$ that satisfies a nice algebraic relation with the transfer matrix. As the notation indicates, the edge operator is defined to satisfy

$$\alpha_j \alpha_{L+1} = -(-1)^{\delta_{jL}} \alpha_{L+1} \alpha_j, \qquad \beta_j \alpha_{L+1} = -\alpha_{L+1} \beta_j, \qquad \left(\alpha_{L+1}\right)^2 = 1, \tag{91}$$

so that

$$A_j\alpha_{L+1} = (-1)^{\delta_{jL}}\alpha_{L+1}A_j\,, \qquad B_j\alpha_{L+1} = \alpha_{L+1}B_j\,, \qquad C_j\alpha_{L+1} = (-1)^{\delta_{jL}}\alpha_{L+1}C_j\,, \qquad (92)$$

with $\delta_{jL}$ the Kronecker delta. One example of an edge operator is $\alpha_{L+1} = X_{L+1}$, but as we will see in (106), it is best to use another one. An immediate consequence of (92) is that

$$\big[H, \alpha_{L+1}\big] = 2\big(A_j + C_j\big)\alpha_{L+1}\,. \qquad (93)$$

In this section we prove that the edge mode satisfies

$$\big[\alpha_{L+1}, T(u)\big] = \frac{u}{2}\Big\{\big[H, \alpha_{L+1}\big], T(u)\Big\}\,. \qquad (94)$$

An immediate consequence of (94) and (76) is

$$T(u)\big(2\alpha_{L+1} + u[H, \alpha_{L+1}]\big)T(-u) = P(u^2)\big(2\alpha_{L+1} - u[H, \alpha_{L+1}]\big)\,, \qquad (95)$$

extending the relation of [10] to our more general transfer matrix.

The proof of (94) is straightforward, requiring only repeatedly using the algebra (22,23,92) along with the recursion relation for the transfer matrix (75). Useful rewritings of the latter are

$$T_j = T_j^B - u\big(T_{j-2}C_j + T_{j-2}^B A_j\big) = T_{j-1} - u\big(T_{j-2}C_j + T_{j-2}^B A_j + T_{j-2}^A B_j\big)\,. \qquad (96)$$

Because $\alpha_{L+1}$ commutes with all transfer matrices $T_j$ with $j < L-1$, we use the last relation and the algebra to simplify the left-hand side of (94) to

$$\big[\alpha_{L+1}, T_L\big] = 2u\big(T_{L-2}^B A_L + T_{L-2}C_L\big)\alpha_{L+1}\,. \qquad (97)$$

The strategy to simplify the right-hand side is to keep using the recursion relation and algebra to remove contributions to $T_L$ that anticommute, finally leaving only commuting bits. Thus

$$\begin{aligned}
\big\{T_L, A_L\alpha_{L+1}\big\} &= \big\{T_{L-1} - u\big(T_{L-2}C_L + T_{L-2}^B A_L + T_{L-2}^A B_L\big), A_L\alpha_{L+1}\big\} \\
&= \big\{T_{L-2} - u\big(T_{L-2}C_L + T_{L-2}^A B_L\big), A_L\alpha_{L+1}\big\} \\
&= \big\{T_{L-2}^B - u\big(T_{L-2}^B C_L + T_{L-3}B_L\big), A_L\alpha_{L+1}\big\} \\
&= 2\big(T_{L-2}^B - u\big(T_{L-2}^B C_L + T_{L-3}B_L\big)\big)A_L\alpha_{L+1}\,,
\end{aligned}$$

and

$$\begin{aligned}
\big\{T_L, C_L\alpha_{L+1}\big\} &= \big\{T_{L-1} - u\big(T_{L-2}C_L + T_{L-2}^B A_L + T_{L-2}^A B_L\big), C_L\alpha_{L+1}\big\} \\
&= \big\{T_{L-2} - uT_{L-3}C_{L-1} - uT_{L-2}^B A_L, C_L\alpha_{L+1}\big\} \\
&= 2\big(T_{L-2} - u\big(T_{L-3}C_{L-1} + T_{L-2}^B A_L\big)\big)C_L\alpha_{L+1}\,.
\end{aligned}$$

Then because $\{A_L, C_L\} = 0$ we have

$$\big\{T_L, \big(A_L + C_L\big)\alpha_{L+1}\big\} = 2\big(T_{L-2}^B A_L + T_{L-2}C_L - uT_{L-3}\big(B_L A_L + C_{L-1}C_L\big)\alpha_{L+1}\big)\,.$$

Using the forbidden-pair relation (21) along with (97) and (93) yields (94).

## 5.2 The raising/lowering operators and their algebra

The raising and lowering operators are now easy to construct following [10]. They are defined in terms of the edge mode and the transfer matrix at the roots of the polynomial $P(u^2)$ defined by (77). There are $2r_{max} = 2\lfloor(L+1)/2\rfloor$ such roots, which we write as $\pm u_k$ with $k = 1, 2, \ldots r_{\max}$. These roots are non-zero, since by construction $P(0) = 1$, and are real for $a_j$, $b_j$ real. The latter statement is not immediately apparent, but follows from the subsequent analysis. We then have $k$ raising and $k$ lowering operators defined by

$$\Psi_{\pm k} = \frac{1}{N_k} T(\mp u_k)\alpha_{L+1} T(\pm u_k),\tag{98}$$

where $N_k$ is a normalisation defined below. Then (95) applied at $u = \pm u_k$ along with the fact that $H$ commutes with $T(u)$ yield

$$\left[H, \Psi_{\pm k}\right] = \pm\frac{2}{u_k}\Psi_{\pm k}.\tag{99}$$

The relation (99) is the canonical definition of a raising (+) and lowering (-) operator. Acting with $\Psi_{+k}$ on an eigenstate of $H$ either annihilates the state or gives another state with energy raised by $2/u_k$. Likewise, acting with $\Psi_{-k}$ either annihilates or lowers the eigenvalue. The connection to the free-fermion spectrum (1) is obvious once we identify

$$\epsilon_k = \frac{1}{u_k}.\tag{100}$$

This spectrum is then the simplest possibility consistent with (99).

To establish that the set of raising/lowering operators is complete and that the spectrum is precisely that in (1), we need to derive several properties of the $\Psi_{\pm k}$. Luckily, we do not need to do much additional work, now that we have dealt with the complications of having a frustration graph with claws. We have derived (48), (76) and (95), which correspond respectively to equations (2.10), (2.16) and (3.11) of [10] and Lemmas 1, 2 and 3 in [11]. The needed properties of the raising/lowering operators requires only this information, and so we can utilise the subsequent proofs wholesale. We thus do not repeat them and just give the results.

The first property needed is that the $\Psi_{\pm k}$ satisfy a Clifford algebra when their normalisation is chosen appropriately:

$$\left\{\Psi_l, \Psi_{l'}\right\} = \delta_{l,-l'}, \qquad (N_k)^2 = -8u_k P_L^B\left(u_k^2\right)\partial_u P_L(u^2)\big|_{u=u_k}.\tag{101}$$

The proof of these relations is found in the analysis leading to equations (1.3) and (3.20) of [10], or equivalently from Lemma 4 of [11]. Because $\{\Psi_k, \Psi_{-k}\} = 1$, no state can be annihilated by both of $\Psi_k$, $\Psi_{k'}$. Moreover, $(\Psi_l)^2 = 0$ means that no raising or lowering operator can be applied twice. The energy spectrum (1) remains the simplest possibility consistent with these facts.

One more result establishes that the spectrum of $H$ is precisely that of (1) with $M = r_{\max} = \lfloor(L+1)/2\rfloor$. Using the proof leading to equation (3.21) of [10], we can rewrite the Hamiltonian $H = H^{(1)}$ and the local conserved charges as bilinears in the raising and lowering operators:

$$H^{(2n+1)} = \sum_{k=1}^{r_{\max}} u_k^{-(2n+1)}\left[\Psi_k, \Psi_{-k}\right],\tag{102}$$

where $m$ is any non-negative integer. Given the $\Psi_{\pm k}$ satisfy Clifford algebra (101) and the raising/lowering properties from (99), the spectrum of $H = H^{(1)}$ with fixed boundary conditions

must be included in the eigenvalues given in (1), once we identify $\epsilon_k = 1/u_k$. Likewise, all eigenvalues of $H^{(2n+1)}$ are contained in

$$E^{(2n+1)} = \pm\epsilon_1^{2n+1} \pm \epsilon_2^{2n+1} \pm \cdots \pm \epsilon_{r_{\max}}^{2n+1}. \tag{103}$$

The operators $H^{(2n)}$ are proportional to the identity, with eigenvalues $\sum_k (\epsilon_k)^{2n}$.

### 5.3 The spectrum and its degeneracies

While our analysis leading to (103) shows that all energies must be of this form, it does not guarantee that all such energies appear in the spectrum of $H$ with a given fixed boundary condition. The reason is that we have not checked that the raising/lowering operators we have constructed leave the boundary conditions invariant. Here we show that for even $L$, the spectrum includes all the energies in (1), but that for odd $L$ only half of them appear. We then show each energy has an exponentially large degeneracy.

It is first useful to construct a set of operators that anticommute with $H$ and square to 1:

$$\mathcal{C}_l \equiv X_l Y_{l+1} X_{l+4} Y_{l+5} X_{l+8} Y_{l+9} \ldots, \qquad \mathcal{C}_3 \equiv Y_0 X_3 Y_4 X_7 Y_8 X_{11} \ldots, \tag{104}$$

where $l = 0, 1, 2$ and the products truncate at index $L+1$ for fixed boundary conditions. It is easy to check that $\mathcal{C}_l h_m = -h_m \mathcal{C}_l$ for any $l, m$. However, depending on the value $L$, the $\mathcal{C}_l$ may change the fixed boundary conditions. For $L$ a multiple of 4, only $\mathcal{C}_2$ leaves the boundary spins unchanged. For $L = 4n + 2$ with $n$ a non-negative integer, $\mathcal{C}_1$ works, while for $L = 4n + 3$ both $\mathcal{C}_1$ and $\mathcal{C}_2$ work. However, for $L = 4n + 1$ with $n$ a non-negative integer, all the $\mathcal{C}_l$ change the boundary conditions. The spectrum therefore is invariant under $H \to -H$ for $L \neq 4n + 1$.

Products $\mathcal{C}_l \mathcal{C}_{l'}$ then yield discrete symmetries. For the fixed boundary conditions of interest here, symmetries must leave the spins at sites 0 and $L+1$ unchanged. For any length $L$, the $\mathbb{Z}_2$ symmetry from (13) is $(-1)^F \propto \mathcal{C}_0 \mathcal{C}_1 \mathcal{C}_2 \mathcal{C}_3$. The operator

$$R = Z_0 Y_1 X_3 Z_4 Y_5 X_7 \ldots \propto \mathcal{C}_0 \mathcal{C}_3, \tag{105}$$

then generates another $\mathbb{Z}_2$ symmetry for $L = 4n + 1$, $4n + 3$. Thus for $L$ odd, our Hamiltonian with fixed boundary conditions has a $\mathbb{Z}_2 \times \mathbb{Z}_2$ symmetry, while for $L$ even the discrete symmetry is only $\mathbb{Z}_2$, generated by $(-1)^F$.

We now can complete our understanding the spectrum. The raising/lowering operators $\Psi_{\pm k}$ from (98) include the operator $\alpha_{L+1}$ satisfying the algebra (91). The simplest choice $\alpha_{L+1} = X_{L+1}$ does not preserve the fixed boundary conditions, as it flips the spin at site $L+1$. However, we can use the operator $R$ defined in (105) to make a different choice that leaves the boundary conditions invariant. Namely, note that for $L$ even, $R$ commutes with each term in $H$ and hence $T(u)$, but flips the spin at $L+1$. Thus if we instead take

$$\alpha_{L+1} = i^{L/2+1} X_{L+1} R, \tag{106}$$

the ensuing raising/lowering operators preserve the fixed boundary conditions when $L$ is even. Since using $\alpha_{L+1} = X_{L+1}$ instead gives raising/lowering operators that change the boundary spin, the spectrum for even $L$ is independent of which fixed boundary conditions are taken.

Acting with all $r_{\max} = L/2$ of the $\Psi_{\pm k}$ thus shows that all $2^{L/2}$ energies in (1) or (103) occur in the spectrum for $L$ even. For $L$ odd, however, they do not, because any valid choice for $\alpha_{L+1}$ flips a boundary spin. Thus acting with any $\Psi_{\pm k}$ changes the boundary conditions $(Z_0, Z_{L+1}) \leftrightarrow (Z_0, -Z_{L+1})$ for $L$ odd, but a with two of them, however, restores the original boundary conditions. Thus for a given fixed boundary condition $(++)$, only half the spectrum (1) occurs, with the other half occurring when the boundary condition is $(+-)$.

This behaviour is easy to illustrate by considering a small number of sites. For $L = 1$, the Hamiltonian is simply $a_1 b_1 Z_0 Z_2$, while the two $\mathbb{Z}_2$ symmetries are $(-1)^F = Z_1$ and $R = X_1$. The energies come from the roots $\pm u_1 = \pm(a_1 b_1)^{-1}$ of the polynomial $P_1(u^2) = 1 - (a_1 b_1 u)^2$. The possibilities (1) for the energies are thus $E = \pm a_1 b_1$. For $(\pm\pm)$ fixed boundary conditions, $E = a_1 b_1$ for either state at site 1, as required from the discrete symmetry $R$. For $(\pm\mp)$ fixed boundary conditions, $E = -a_1 b_1$. Each state is doubly degenerate.

For $L = 2$, $r_{\max} = 1$, so the roots of the polynomial $P_2(u^2)$ and the energies are

$$E = \pm u_1^{-1} = \pm\sqrt{(a_1 b_1)^2 + (a_2 b_2)^2 + (a_1 a_2)^2 + (b_1 b_2)^2}\,. \tag{107}$$

The Hamiltonian for fixed boundary conditions is

$$H\big|_{L=2} = a_1 b_1 Z_0 Z_2 + a_2 b_2 Z_1 Z_3 + a_1 a_2 X_1 X_2 X_3 + b_1 b_2 Z_0 Y_1 Y_2\,. \tag{108}$$

By using the symmetry $(-1)^F$, this Hamiltonian splits into two blocks

$$\begin{pmatrix} a_1 b_1 Z_0 + a_2 b_2 Z_3 & a_1 a_2 Z_3 - b_1 b_2 Z_0 \\ a_1 a_2 Z_3 - b_1 b_2 Z_0 & -a_1 b_1 Z_0 - a_2 b_2 Z_3 \end{pmatrix}, \quad \text{and} \quad \begin{pmatrix} -a_1 b_1 Z_0 + a_2 b_2 Z_3 & a_1 a_2 Z_3 + b_1 b_2 Z_0 \\ a_1 a_2 Z_3 + b_1 b_2 Z_0 & a_1 b_1 Z_0 - a_2 b_2 Z_3 \end{pmatrix},$$

that indeed both have eigenvalues (107) for any of the four fixed boundary conditions. The spectrum is thus doubly degenerate.

The $L = 3$ case is also illuminating. Here $r_{\max} = 2$ so

$$P_3(u^2) = 1 - u^2\big((a_1 b_1)^2 + (a_2 b_2)^2 + (a_3 b_3)^2 + (a_1 a_2)^2 + (a_2 a_3)^2 + (b_1 b_2)^2 + (b_2 b_3)^2\big)$$
$$+ u^4 (a_1 b_1 a_3 b_3)^2\,,$$

has two pairs of roots $\pm u_1$ and $\pm u_2$, yielding four distinct energies. The energies still can be worked out by hand: because the symmetry here is $\mathbb{Z}_2 \times \mathbb{Z}_2$ ($R = Z_0 Y_1 X_3 Z_4$), the Hamiltonian can be decomposed into four 2×2 blocks. One finds that all four blocks have the same eigenvalues, so each energy is four-fold degenerate. Therefore only two distinct energies occur for a given fixed boundary condition, despite there being four possible energies. Explicit calculation shows that for $(++)$ or $(--)$ boundary conditions, the two energies are $E = \pm(\epsilon_1 + \epsilon_2)$, while for $(+-)$ or $(-+)$, they are of the form $E = \pm(\epsilon_1 - \epsilon_2)$, where $\epsilon_1$ and $\epsilon_2$ indeed are the inverses of the roots of $P_3(u^2)$. Acting on the state with $E = \epsilon_1 + \epsilon_2$ with both lowering operators $\Psi_{-1}\Psi_{-2}$ does preserve the $(++)$ or $(--)$ boundary condition and gives the state with energy $E = -\epsilon_1 - \epsilon_2$. The spectrum is invariant under $E \to -E$ as it must be for $L = 3$.

Our formalism of course works for any $L$. Since $r_{\max} = \lfloor (L+1)/2 \rfloor$, the number of distinct energies grows much slower than the number of eigenvalues of $H$. Since the Hamiltonian is written as (102), it immediately follows that each energy is exponentially degenerate, with the identical degeneracy for all. The analysis of this subsection gives this degeneracy exactly.

Table 1: Properties of spectrum for different $L$ for fixed boundary conditions.

|  | $L = 4n$ | $L = 4n + 1$ | $L = 4n + 2$ | $L = 4n + 3$ |
|---|---|---|---|---|
| invariant under $E \to -E$? | yes | no | yes | yes |
| discrete symmetry | $\mathbb{Z}_2$ | $\mathbb{Z}_2 \times \mathbb{Z}_2$ | $\mathbb{Z}_2$ | $\mathbb{Z}_2 \times \mathbb{Z}_2$ |
| # energy levels $r_{\max}$ | $2n$ | $2n + 1$ | $2n + 1$ | $2n + 2$ |
| Action of $\Psi_{\pm k}$ on b.c. | invariant | $(++) \leftrightarrow (+-)$ | invariant | $(++) \leftrightarrow (+-)$ |
| # distinct energies | $2^{2n}$ | $2^{2n}$ | $2^{2n+1}$ | $2^{2n+1}$ |
| degeneracy of each state | $2^{2n}$ | $2^{2n+1}$ | $2^{2n+1}$ | $2^{2n+2}$ |

We collect these numbers along with a summary of all the properties of the spectrum in Table 1. This table applies to the case where all $a_j$ and $b_j$ are non-vanishing. In the ffd case $b_j = 0$ (or equivalently instead all $a_j = 0$), $r_{max}$ becomes the smaller value $\lfloor (L-1)/3 \rfloor$ and the degeneracies here are larger.

## 6   Conclusion

We have computed the exact free-fermion spectrum of family of spin chains that does not have a Jordan-Wigner transformation to fermion bilinears. This family unifies two previously known but seemingly distinct examples [9, 10] of such behaviour into a common framework. Our results allow for certain spatially varying couplings, and demonstrate that integrability if not the free-fermion spectrum still holds with periodic boundary conditions. The same techniques can also be applied directly to the Cooper-pair chain of [5], yielding the same spectrum up to degeneracies. Precisely, one unitarily transforms (see eqn. (18) in [21]) the Cooper-pair chain to an equivalent model of coupled bosonic and fermionic chains [22]. The generators of the resulting Hamiltonian obey the same algebra as the DFNR model. We thus can deform the Cooper-pair model as described above, and so find another family of models with free-fermion spectra.

Our analysis required extending the approach of [10,11] further, as the interactions here do not obey the conditions needed to utilise the graphical results of [11, 12]. The relation (21) between Hamiltonian generators proved crucial in this extension. Our results thus suggest that this graphical approach can be extended further. Another potential generalisation is in the direction of "free parafermions", which can be solved [23] and generalised [13, 24] in a very analogous fashion. As these models involve $n$-state systems (i.e. qudits) and have a $\mathbb{Z}_n$ symmetry, the graphs needed here presumably would need to be directional. Also worth pointing out is that since our methods allow for spatially varying couplings, one has some headway in analysing these models with random couplings [25].

The connection with supersymmetry is rather mysterious. Here it led to a non-abelian symmetry algebra that (at least in part) explains the large degeneracies. Is supersymmetry required for models for our methods to be applicable, or are other consistent non-abelian symmetry algebras possible? Another mystery is how the degeneracies split when periodic boundary conditions are imposed, e.g. if there is a second dynamical critical exponent describing these modes.

A more general mystery is how our approach relates to the traditional approach to integrability using Lax operators. We originally found our Hamiltonian in its translationally invariant form by exploring the integrability of medium-range spin chains as discussed in section 2.1, constructing a Lax operator that leads to commuting transfer matrices for translationally invariant couplings. However, it is not apparent how to analyse the spatially varying couplings in $H$ or $H_p$ in a traditional setup. Moreover, the transfer matrices coming from the Lax operators are not the same as our transfer matrix $T(u)$, even for uniform couplings. Indeed, the local conserved charge $\mathcal{H}_2$ from (6) derived from $H_{b,p}$ does not appear to follow from $T(u)$: as explained in (34) and near (38), the charge $Q^{(2)}$ stemming from bilinears in the Hamiltonian generators is trivially related to $H^2$. We plan to return to these issues and the other connections with integrability in the future [20].

## Acknowledgments

We thank T. Gombor and E. Ilievski for useful discussions.

**Funding information**   This work of P.F. was supported in part by EPSRC grant EP/S020527/1.

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
