# Peer review of "Free fermions beyond Jordan and Wigner"

_SciPost Physics, doi:SciPost Phys. 16, 102 (2024)_

## Round 2 · Referee Report · Anonymous (Referee 1) · 2024-1-12

Strengths

1-Relevant new materials and methods to the area of exact integrable models 2- Possible extensions and applications of the ideas presented in the paper 2-Clear presentation

Report

Most of the known quantum chains with a free fermionic spectrum are the ones
that after a Jordan-Wigner transformation brings the quantum chain in a sum
of bilinear fermion operators. Some exceptions however appear in the
literature. These are models that although not bilinear after the
Jordan-Wigner transformation, also show a free-fermion eigenspectra.
Two examples of models with three-spin multispin interactions are given by the
Hamiltonian DFNR (ref. [9]) and FFD (ref. [10]). The connection between these
quantum chains is not known. In the present paper the authors introduce an
extended model that interpolate both Hamiltonians continuously, clarifying the
free-fermion nature of these exceptional models.

The proof of their results were obtained by exploiting some general results
(refs.[11-12], for free-fermion defined in graphs. They also show that the
free-fermionic excitations are obtained from the roots of a polynomial.

The paper is well written and the results are significant for the area of
exact integrable models. We recommend the publication.

Suggestion: The title seems misleading, perhaps "Free fermions models not solved by Jordan-Wigner transformation", or something on this line ...

Question: Is it possible to write in the paper the general recursion relations
for the polynomial p(u)?, from (4.2) and (4.3)? In the positive case it will
be interesting to show this relation.

---

## Round 2 · Referee Report · Bernard Nienhuis (Referee 2) · 2024-1-22

Strengths

1 A very detailed and complete analysis of a new quantum chain.

2 The place in the brief history of the subject is well documented in the text.

Weaknesses

The paper is highly technical. It takes a lot of stamina to read it through, and diagonal reading is virtually impossible.

Report

This interesting paper discusses a new quantum chain, with a Hamiltonian trilinear in Pauli operators. It has the property that its spectrum is that of free fermions, in which the fermion energies are the roots of a polynomial. Remarkably it combines this property with supersymmetry and has two supersymmetry generators. The model shares these properties with several others in the literature, and in fact interpolates between two of them.
The model is analysed extensively, resulting in a list of local and non-local conserved quantities, as well as the spectrum. The analysis is highly technical, but to a large extent self contained.

In spite of the technicality in most places it is clear where the text is heading. I consider it very positive that the connection with related models already in the literature is msde very explicit.

In my opinion this paper brings together several research lines in mathematical statistical physics: integrability, supersymmetry, free fermions, and possibly Cooper pairs.
It is written clearly, and the derivations can be reproduced, with only small number of necessary references.
I recommend publication in SciPost.

Requested changes

1) In the title: Aside from the ugly "with no" I find the title misleading. The Jordan-Wigner transformation is quite explicit in eq. (2.5). What is different from the well known applications of this transformation is that the Hamiltonian is not bilinear in the resulting (Majorana) fermion operators. What I find the most important properties of the paper is that it deals with (i) free fermions, (ii) supersymmetry, (iii) explicit spectrum and (iv) explicit conserved quantities, local and non-local.

2) Between eq. (2.2) and (2.3) it says: "If the Hamiltonian is derived via (2.1) then ${\cal H}_2$ is always the next local charge..." What is the meaning here of always? Always under what variation? And what is kept fixed; not only (2.1) I am convinced, but that is what is implied. Something similar plays in page 4 line 6: in "for all known solutions of (2.3) they do exist", the side conditions are not clear, presumably (2.2) and the embedded expression for $\cal H$.

3) Sec 2.4 is very casual about the Hamiltonian being equal to the square of two supersymmetry generators up to a different additive constant, stating that the larger constant is less a useful bound (I would say more useful). It does not spend a word on the fact that the lower bound of the two bounds can not be reached, which implies that there are no supersymmetry singlets with respect to the corresponding generator. And if the effective groundstates are singlets with respect to one of the generators (the one with the larger constant), they must form doublets with respect to the other generator.

---

## Round 3 · Author Response

We thank the referees for the comments. We have made several minor changes, and hope that it is sufficient.

We have a comment regarding the "weakness" that Referee 2 stated ("The paper is highly technical. It takes a lot of stamina to read it through, and diagonal reading is virtually impossible.")

We recognise that we wrote a paper that does not allow for diagonal reading, but do not believe it is a weakness. We believe our results are interesting enough to make them worthy of a careful proof, and that we would not be serving readers well by omitting them. Our introduction and conclusion state the results in a clear and concise fashion, and we hope that this suffices for casual readers. In general we do not consider this style of theoretical physics to be weak, and hope that the SciPost editors agree.

---

## Round 3 · List of Changes

Regarding the requested changes from Referee 2:

1 (also from Referee 1). We note that while there are Jordan-Wigner fermions in the paper, they are not the free-fermion operators (as the original title indicated). We modified the title to avoid this confusion.

  1. We clarified this sentence, without giving all details of the construction. This point is not central to the rest of the paper, and so we did not specify all the mathematical details, for example the regularity condition in question. We refer to the work [14] for details.

  2. We note that the larger constant is less useful because Q^2 = const + H. Since eigenvalues of Q are non-negative, - const is a lower bound on the spectrum of H. We clarified the text, and also added a sentence on forming doublets.

Regarding the question from Referee 1:

There is indeed an analogous recursion relation for the polynomials, and we have added the relation to the paper in equation (4.6).

---

## Editorial Decision

published